# Survival Outcomes of Patients with Mantle Cell Lymphoma: A Retrospective, 15-Year, Real-Life Study

Emanuele Cencini [1,*], Natale Calomino [2], Marta Franceschini [1], Andreea Dragomir [1], Sara Fredducci [1], Beatrice Esposito Vangone [1], Giulia Lucco Navei [1], Alberto Fabbri [1] and Monica Bocchia [1]

1   Unit of Hematology, Azienda Ospedaliera Universitaria Senese, University of Siena, 53100 Siena, Italy; martafranceschini@student.unisi.it (M.F.); andreead020995@student.unisi.it (A.D.); sara.fredducci@student.unisi.it (S.F.); b.espositovangone@student.unisi.it (B.E.V.); giulia.lucconavei@student.unisi.it (G.L.N.); fabbri7@unisi.it (A.F.); bocchia@unisi.it (M.B.)

2   Unit of General Surgery and Surgical Oncology, Department of Medicine, Surgery and Neurosciences, University of Siena, 53100 Siena, Italy; natale.calomino@unisi.it

*   Correspondence: emanuele.cencini@ao-siena.toscana.it; Tel.: +39-0577586798

**Abstract:** Mantle cell lymphoma (MCL) prognosis has significantly improved in recent years; however, the possible survival benefit of new treatment options should be evaluated outside of clinical trials. We investigated 73 consecutive MCL patients managed from 2006 to 2020. For younger patients <65 years old, the median PFS was 72 months and we reported a 2-year, 5-year, and 10-year PFS of 73%, 62%, and 41%; median OS was not reached and we reported a 2-year, 5-year, and 10-year OS of 88%, 82%, and 66%. For patients aged 75 years or older, the median PFS was 36 months and we reported a 2-year, 5-year, and 10-year PFS of 52%, 37%, and 37%; median OS was not reached and we reported a 2-year, 5-year, and 10-year OS of 72%, 55%, and 55%. The median PFS was significantly reduced for patients treated between 2006 and 2010 compared to patients treated between 2011 and 2015 ($p = 0.04$). Interestingly, there was a trend towards improved OS for patients treated between 2016 and 2020 compared to between 2006 and 2010 and between 2011 and 2015 (5-year OS was 91%, 44%, and 33%). These findings could be due to the introduction of BR as a first-line regimen for elderly patients and to the introduction of ibrutinib as a second-line regimen.

**Keywords:** mantle cell lymphoma; chemoimmunotherapy; target therapy; survival

## 1. Introduction

Mantle cell lymphoma (MCL) represents a rare subtype of non-Hodgkin lymphoma (NHL), characterized by cyclin D1 overexpression, that is associated with t(11;14)(q13;q32) translocation [1–4]. MCL incidence has increased over the last few years, especially among elderly patients [2,4]. MCL is commonly diagnosed as advanced-stage, with frequent nodal, extranodal, and bone marrow involvement, and it is divided into classical, pleomorphic, and blastoid variants, with an aggressive disease course in most cases [1–4]. However, 10–20% of patients are characterized by a leukemic, non-nodal (LNN) presentation, with more indolent clinical behavior and prolonged survival [4–6]. The MCL international prognostic index (MIPI), together with Ki-67 expression via immunohistochemistry, represents the most used prognostic tool used at diagnosis [7,8]. Moreover, the absence of SOX-11 overexpression is associated with LNN presentation [4–6].

For younger patients, first-line therapy is represented by high-dose (HD) cytarabine-containing regimens, followed by autologous stem-cell transplantation (ASCT) as well as consolidation and rituximab maintenance [9,10]. For patients who are ineligible for HD cytarabine and ASCT, various regimens have been investigated with promising results, such as rituximab and bendamustine (BR), rituximab, bendamustine and an intermediate dose of cytarabine (R-BAC), and rituximab in association with bortezomib, cyclophosphamide, doxorubicin, and prednisone (VR-CAP) [11–14]. The R-BAC regimen has been supported

by preclinical studies, in which bendamustine and cytarabine have shown synergistic mechanisms of action; however, this high antineoplastic activity is counterbalanced by greater toxicity compared to BR [11–13]. Despite improved overall survival (OS) during recent years, due to deeper remissions after first-line therapy, as well as more efficacious salvage regimens, survival curves have not reached a plateau in most clinical trials and late relapses have been frequently reported [15]. Relapsed or refractory (R/R) MCL cases have a dismal prognosis, with a shortened duration of complete response (CR) after subsequent regimens [4]. Many novel agents have been approved for R/R MCL cases, including mTOR inhibitors, immunomodulatory drugs, proteasome inhibitors, and Bruton tyrosine kinase (BTK) inhibitors [16–19]. In a phase II study, bortezomib was administered to 155 R/R MCL patients, with an overall response rate (ORR) of 33%; the median duration of response and time to progression were 9.2 and 6.2 months, respectively [16]. In both clinical trials and real-life experiences, lenalidomide showed encouraging PFS, even for patients with MCL who relapsed after bortezomib [17]. In a multicenter, phase III study, patients treated with temsirolimus had longer PFS than those treated with the investigator's choice of regimen (the median PFS was 4.8 and 1.9 months, respectively) [18].

Ibrutinib is an oral, covalent BTK inhibitor, which demonstrated encouraging long-term efficacy in several NHL subtypes, including R/R MCL [19–23]. Ibrutinib showed prolonged progression-free survival (PFS) and OS with mild toxicity, especially as a second-line regimen; however, survival was reduced for patients with chemo-refractory MCL and/or TP53 mutations [24,25]. For patients failing ibrutinib, overall prognosis is extremely poor after ibrutinib failure, with an unsatisfactory response to subsequent regimens [26]. In a retrospective study across 15 international sites, in which 114 patients were enrolled, median OS after ibrutinib failure was only 2.9 months [26]. In this setting, promising results were recently obtained by chimeric antigen receptor (CAR) T-cells [27,28]. Brexucabtagene autoleucel is a CD19-directed CAR T-cell therapy approved for R/R MCL after two previous regimens; 189 cases who underwent leukapheresis were enrolled in a real-life study. ORR and CR were 90% and 82%, respectively, with an estimated 12-month PFS of 59% after a median follow-up of 14.3 months [27,28]. However, this approach is not consistently feasible, due to restrictive eligibility criteria, progressive disease (PD), or death while waiting for manufacturing [27,28]. Furthermore, major treatment toxicities of CAR T-cell therapy include cytokine release syndrome, immune effector cell-associated neurotoxicity syndrome, prolonged cytopenia, and infections [27,28].

Remarkably, patients who are enrolled in clinical trials are often fit and healthier compared to real-life populations; as a result, many elderly MCL patients were not included in these studies. Despite the large amount of newly diagnosed or R/R MCL cases requiring treatment, there are limited long-term follow-up experiences in a non-trial setting [29–31]. Hence, at our institution, we retrospectively investigated the long-term efficacy and toxicity of MCL patients receiving available treatment options in daily clinical practice.

## 2. Materials and Methods

### 2.1. Study Population

From 2006 to 2020, 73 consecutive newly diagnosed or R/R MCL patients were enrolled in this observational, real-world, monocenter study.

Diagnosis was made according to the World Health Organization Classification of Tumours of Haematopoietic and Lymphoid Tissues (WHO) 2008 and 2016 classifications [32]. All alive patients signed written informed consent and the study was approved in accordance with Institutional Review Board requirements and the Declaration of Helsinki and its amendments (Comitato Etico Regionale per la Sperimentazione Clinica della Regione Toscana Sezione: AREA VASTA SUD EST, protocol approval on 15 October 2023, protocol code SI_MCL, protocol number 25529). Regarding the participants who died, consent was considered to be acquired by Italian law, according to Authorization n.9/2016—General Authorization to Process Personal Data for Scientific Research (date of authorization was 15 December 2016). The medical records were reviewed to calculate simplified MIPI scores

(sMIPIs). Baseline patient information, the treatment lines administered, and survival data were collected in an electronic database.

Patients were included if they had an MCL diagnosis according to WHO classification, were ≥18 years old at diagnosis, and received at least 1 cycle of systemic therapy according to the prescribing information approved by the "Agenzia Italiana del Farmaco", and if there were available data regarding treatment response and long-term toxicity.

As regards the treatment criteria, at our institution, we have used the European Society of Medical Oncology (ESMO) guidelines since 2013, the year in which they were published [4,33]. In previous years, since MCL therapy had not yet been standardized, we adopted expert recommendations [34]. Overall, due to MCL being an aggressive disease, our policy was to treat all patients at diagnosis, except those with an LNN presentation [4–6]. Patients who received at least one cycle of therapy were considered to be evaluable for safety analysis.

The primary efficacy outcomes were the long-term OS and PFS, while response to treatment, long-term toxicity, second primary malignancies (SPMs), and causes of death were secondary end-points. We also investigated the potential predictive variables associated with response to treatment, PFS, and OS. Survival analysis was reported according to intention-to-treat. Safety was investigated in terms of SPM or any other toxicity reported during the long-term follow-up period.

## 2.2. Treatment Regimens

All patients were treated in daily clinical practice; we included both those who received regimens with curative intent and candidates for reduced intensity or palliative therapies. Remarkably, all but one patient received rituximab within the first-line treatment but none of them received rituximab as maintenance therapy. The reason for the lack of use of rituximab as a maintenance therapy after a first-line regimen for MCL patients was that it was included in the list of reimbursed drugs, according to Italian law no. 648/1996, on 22 May 2023. Second-line regimens included chemoimmunotherapy, ibrutinib, lenalidomide, and bortezomib; all therapies were administered following the indications currently approved in Italy. Details regarding subsequent treatment strategies for R/R patients are beyond the scope of this study.

Lamivudine prophylaxis was given to HBV-positive patients from the beginning to at least 12 months after treatment. Direct-acting antivirals were administered to HCV-positive cases, concurrently or subsequently to antilymphoma therapy [35]. Anti-microbial prophylaxis for Pneumocystis Jirovecii pneumonia with trimethoprim-sulfamethoxazole was given to all patients, while erythropoietin-stimulating agents and granulocyte colony-stimulating factor were administered according to the summary of product characteristics.

## 2.3. Statistical Analysis

Clinical and biological characteristics of enrolled patients were analyzed according to descriptive statistics. Categorical variables were investigated using chi-square or Fisher's exact test; Fisher's exact test was preferred when the expected frequency was less than 5. PFS was defined as the time from the first day of treatment until disease progression, relapse, death due to any cause, or last follow-up (censored), whichever occurred first; OS was defined as the time from the first day of therapy until death for any cause or last follow-up (censored). Survival after treatment relapse was assessed from the date in which relapse occurred to the last follow-up for alive patients or death for patients who had died.

Survival analyses were assessed using Kaplan and Meier plots and the log rank test for significant associations; a $p$ value < 0.05 was interpreted as being statistically significant. Our findings are reported as a hazard ratio (HR), with a 95% confidence interval (CI). All statistical analyses were performed using Statistical Software MedCalc, 19.6 (MedCalc Software Ltd., Ostend, Belgium; https://www.medcalc.org (accessed on 20 September 2023)).

## 3. Results

### 3.1. Characteristics of Patients

The clinical characteristics of the whole cohort according to treatment era (2006–2010, 2011–2015, and 2016–2020) are represented in Table 1.

**Table 1.** Characteristics of patients at diagnosis.

| Treatment Era | Entire Cohort n = 73 | 2006–2010 n = 19 | 2011–2015 n = 23 | 2016–2020 n = 31 |
|---|---|---|---|---|
| Age: median [range] | 70 [34–92] | | | |
| <65 | 27/73 (37%) | 6/19 (31.6%) | 8/23 (34.8%) | 13/31 (41.9%) |
| 65–74 | 20/73 (27.4%) | 5/19 (26.3%) | 7/23 (30.4%) | 8/31 (25.8%) |
| ≥75 | 26/73 (35.6%) | 8/19 (42.1%) | 8/23 (34.8%) | 10/31 (32.3%) |
| Male | 51/73 (69.9%) | 13/19 (68.4%) | 16/23 (69.6%) | 22/31 (71%) |
| Blastoid/pleomorphic | 9/73 (12.3%) | 1/19 (5.3%) | 3/23 (13%) | 5/31 (16.1%) |
| Stage | | | | |
| I–II | 11/73 (15.1%) | 2/19 (10.5%) | 4/23 (17.4%) | 5/31 (16.1%) |
| III | 9/73 (12.3%) | 2/19 (10.5%) | 3/23 (13%) | 4/31 (12.9%) |
| IV | 53/73 (72.6%) | 15/19 (79%) | 16/23 (69.6% | 22/31 (71%) |
| B-symptoms | 14/73 (19.2%) | 3/19 (15.8%) | 4/23 (17.4%) | 7/31 (22.6%) |
| sMIPI score | | | | |
| low | 10/73 (13.7%) | 2/19 (10.5%) | 3/23 (13%) | 5/31 (16.1%) |
| intermediate | 21/73 (28.8%) | 6/19 (31.6%) | 7/23 (30.4%) | 8/31 (25.8%) |
| high | 42/73 (57.5%) | 11/19 (57.9%) | 13/23 (56.6%) | 18/31 (58.1%) |
| Elevated LDH | 28/73 (38.4%) | 4/19 (21%) | 10/23 (43.5%) | 14/31 (45.2%) |
| ECOG PS 0–1 | 62/73 (84.9%) | 16/19 (84.2%) | 19/23 (82.6%) | 27/31 (87.1%) |

Abbreviations: sMIPI, simplified mantle cell lymphoma international prognostic index; LDH, lactate dehydrogenase; PS, performance status; ECOG, Eastern Cooperative Oncology Group.

All patients received a physical examination, complete blood cell count, and radiological assessment at baseline and prior to each line of therapy. In the whole cohort, the median age was 70 years (range 34–92 years) and 51/73 (69.9%) were male. There were 62/73 (84.9%) cases diagnosed with advanced-stage disease (9/73, 12.3% stage III and 53/73, 72.6% stage IV) and B-symptoms were present in 14/73 cases (19.2%). The histological subtype was blastoid/pleomorphic in 9/73 cases (12.3%), while 2/73 cases had leukemic, non-nodal MCL (2.7%). The MIPI score was low, intermediate, and high in 10/73 (13.7%), 21/73 (28.8%), and 42/73 (57.5%) cases, respectively. Lactate dehydrogenase (LDH) was elevated in 28/73 cases (38.4%) and ECOG performance status was 0–1 in 62/73 cases (84.9%).

### 3.2. Treatment Regimens

First-line regimens, as represented in Table 2, included BR (26/73 cases), HD cytarabine-based therapies (24/73 cases, 21/24 of which responded to treatment and underwent ASCT), R-BAC (9/73 cases), rituximab and CHOP (cyclophosphamide, doxorubicin, vincristine, and prednisone), or CHOP-like regimens (6/73 cases), fludarabine-containing regimens (5/73 cases, 1/5 with oral low-dose fludarabine and rituximab), and alkylating agents (3/73 cases, 2/3 with rituximab). The total number of patients who received bendamustine was 35/73 (47.9%); it was administered to 16/23 patients (69.6%) and to 19/31 patients (61.3%) between 2011 and 2015 and between 2016 and 2020, respectively. Remarkably, in our cohort, only one case did not receive rituximab, who presented with severe cardiac failure (the New York Heart Association score was 3). Overall, we considered the four cases who received alkylating agents, with or without rituximab, and the case treated with low-dose, oral fludarabine and rituximab as palliation.

**Table 2.** First-line regimens for the entire cohort.

| Treatment | Number of Patients (%) | 2006–2010 *n* = 19 | 2011–2015 *n* = 23 | 2016–2020 *n* = 31 |
|---|---|---|---|---|
| Rituximab-bendamustine | 26/73 (35.6%) | 0 | 9/23 (39.1%) | 17/31 (54.8%) |
| High-dose cytarabine-based therapies | 24/73 (32.9%) | 6/19 (31.6%) | 8/23 (34.8%) | 10/31 (32.3%) |
| Autologous stem-cell transplantation | 21/73 (28.8%) | 6/19 (31.6%) | 6/23 (26%) | 9/31 (29%) |
| Rituximab-bendamustine-cytarabine (R-BAC) | 9/73 (12.3%) | 0 | 5/23 (21.7%) | 4/31 (12.9%) |
| Rituximab-CHOP or CHOP-like regimens | 6/73 (8.2%) | 5/19 (26.3%) | 1/23 (4.4%) | 0 |
| Fludarabine-containing regimens | 5/73 (6.8%) | 5/19 (26.3%) | 0 | 0 |
| Rituximab-alkylating agents | 2/73 (2.8%) | 2/19 (10.5%) | 0 | 0 |
| Alkylating agents | 1/73 (1.4%) | 1/19 (5.3%) | 0 | 0 |

Abbreviations: CHOP: cyclophosphamide, doxorubicin, vincristine, and prednisone.

The treatment response was CR for 51/73 cases (69.8%), partial response for 10/73 cases (13.7%), and stable disease/PD for 12/73 cases (16.5%), respectively.

Second-line regimens were administered to 29 cases and included chemoimmunotherapy (11 cases), ibrutinib (8 cases), bortezomib (5 cases), and lenalidomide (5 cases). Interestingly, two cases experienced central nervous system relapse and received ibrutinib or chemoimmunotherapy (one case each). The detailed subsequent treatment strategies for R/R patients are beyond the scope of this study.

*3.3. Survival Analysis*

For the whole cohort, median PFS was 60 months (95% CI 30–84 months) and we reported a 2-year, 5-year, and 10-year PFS of 63%, 50%, and 32%, respectively (Figure 1a). Median OS was not reached and we reported a 2-year, 5-year, and 10-year OS of 80%, 63%, and 51%, respectively (Figure 1b).

For younger patients <65 years old, median PFS was 72 months (95% CI 30–72 months) and we reported a 2-year, 5-year, and 10-year PFS of 73%, 62%, and 41%, respectively (Figure S1); median OS was not reached and we reported a 2-year, 5-year, and 10-year OS of 88%, 82%, and 66%, respectively (Figure S2). For patients aged between 65 and 74 years, median PFS was 36 months (95% CI 12–84 months) and we reported a 2-year, 5-year, and 10-year PFS of 64%, 47%, and 23%, respectively (Figure S3); median OS was 84 months (95% CI 27–96 months) and we reported a 2-year, 5-year, and 10-year OS of 79%, 52%, and 34%, respectively (Figure S4). For patients aged 75 years or older, median PFS was 36 months (95% CI 12–60 months) and we reported a 2-year, 5-year, and 10-year PFS of 52%, 37%, and 37%, respectively (Figure S5); median OS was not reached and we reported a 2-year, 5-year, and 10-year OS of 72%, 55%, and 55%, respectively (Figure S6).

According to specified treatment periods (2006–2010, 2011–2015, 2016–2020), median PFS was 18 months (95% CI 10–60 months), 72 months (95% CI 36–72 months), and not reached, respectively (Figure 2). Median PFS was significantly reduced for patients treated between 2006 and 2010 compared to patients treated between 2011 and 2015 (HR 2.5716, 95% CI 1.15–5.7, *p* = 0.04) and between 2016 and 2020 (HR 1.6803, 95% CI 0.72–3.88, *p* = 0.1, a trend even if not statistically significant). The median duration of observation for the different treatment periods was 36 months (range 6–180 months), 42 months (range 6–120 months), and 24 months (range 6–72 months), respectively. Median OS was 38 months (95% CI 13–96 months), not reached, and not reached for the three treatment periods, with a trend toward a reduced OS for patients treated between 2006 and 2010, even if not statistically significant (Figure 3).

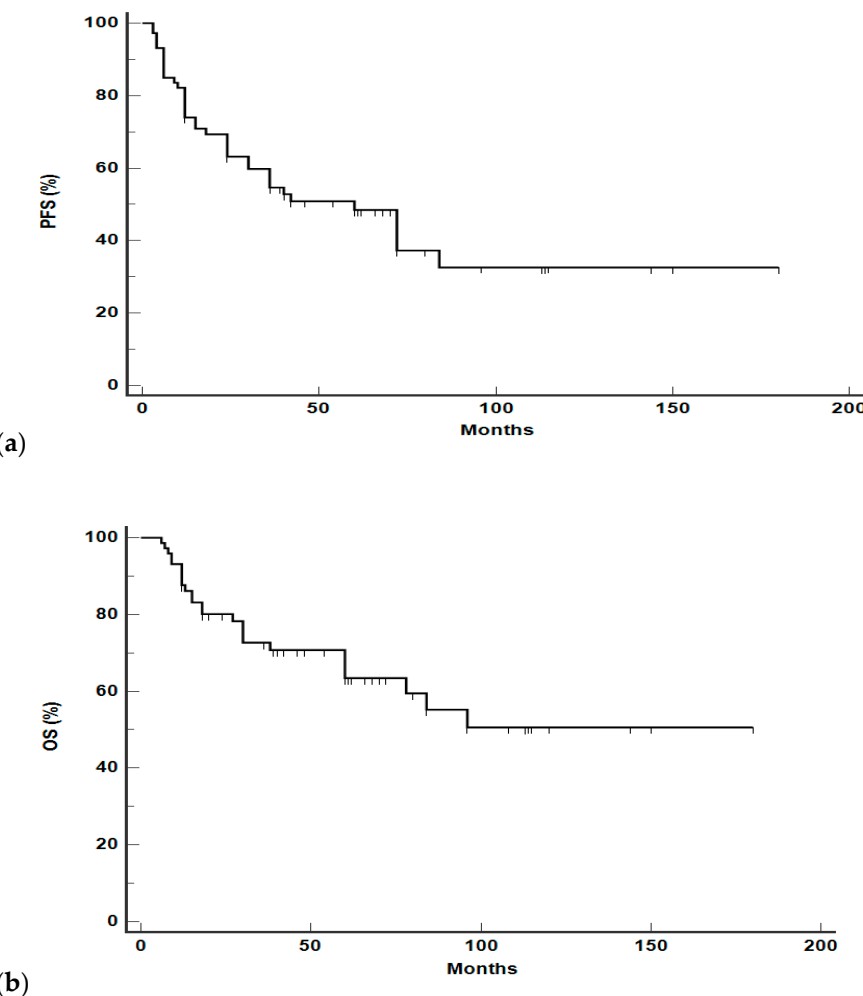

**Figure 1.** PFS (**a**) and OS (**b**) for the entire cohort.

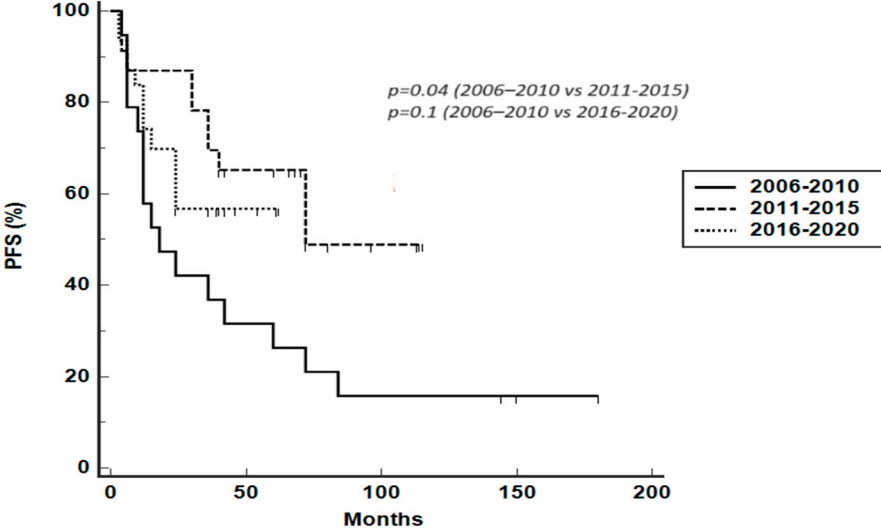

**Figure 2.** PFS according to specified treatment periods (2006–2010, 2011–2015, 2016–2020).

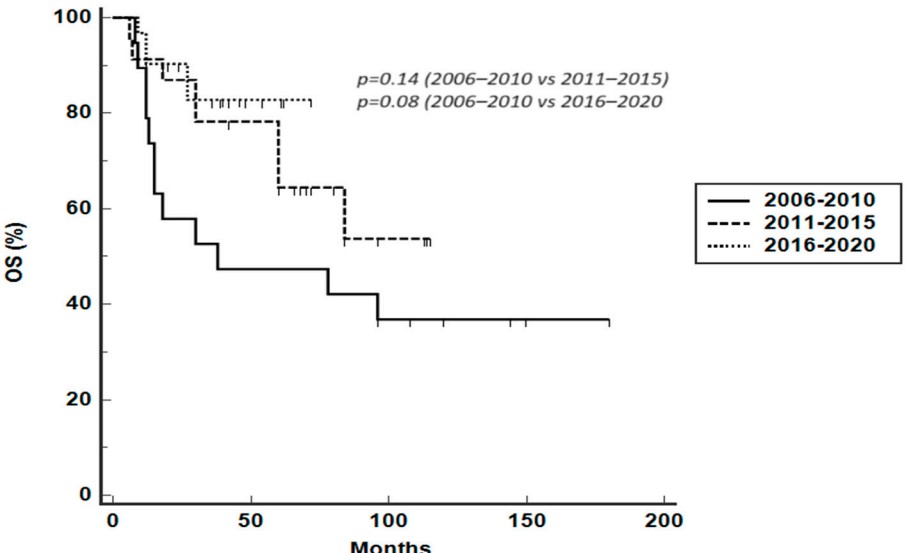

**Figure 3.** OS according to specified treatment periods (2006–2010, 2011–2015, 2016–2020).

When we separately analyzed patients aged 75 years or older, we observed that PFS did not improve across the years (Figure 4a); interestingly, there was a trend towards an improved OS for patients treated between 2016 and 2020 compared to between 2006 and 2010 and between 2011 and 2015 (5-year OS was 91%, 44%, and 33%, respectively), even if it was not statistically significant (Figure 4b).

Univariate analysis showed that MIPI score at diagnosis, advanced-stage disease at diagnosis, gender, histologic subtype, administration of intensive front-line regimen, and ASCT were not associated with PFS and OS.

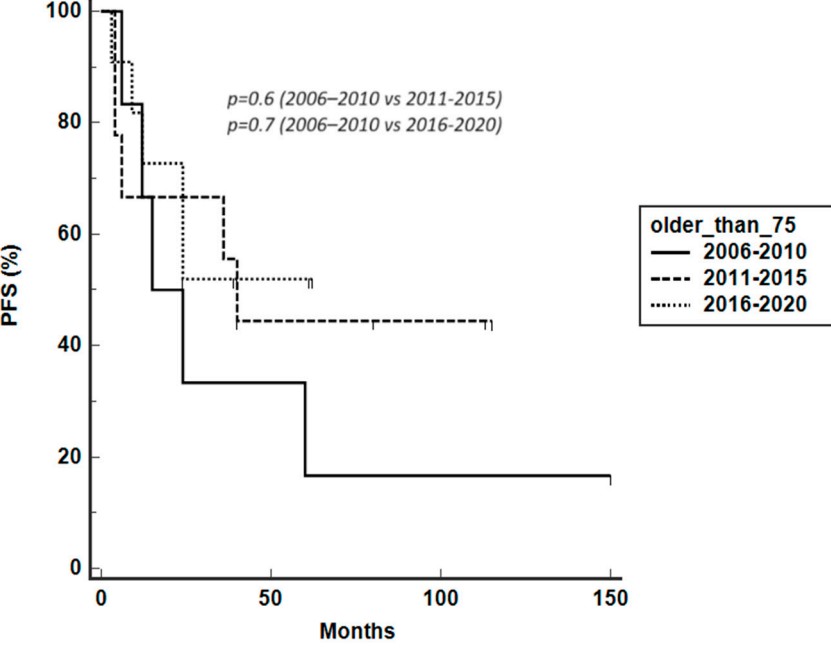

(**a**)

**Figure 4.** *Cont*.

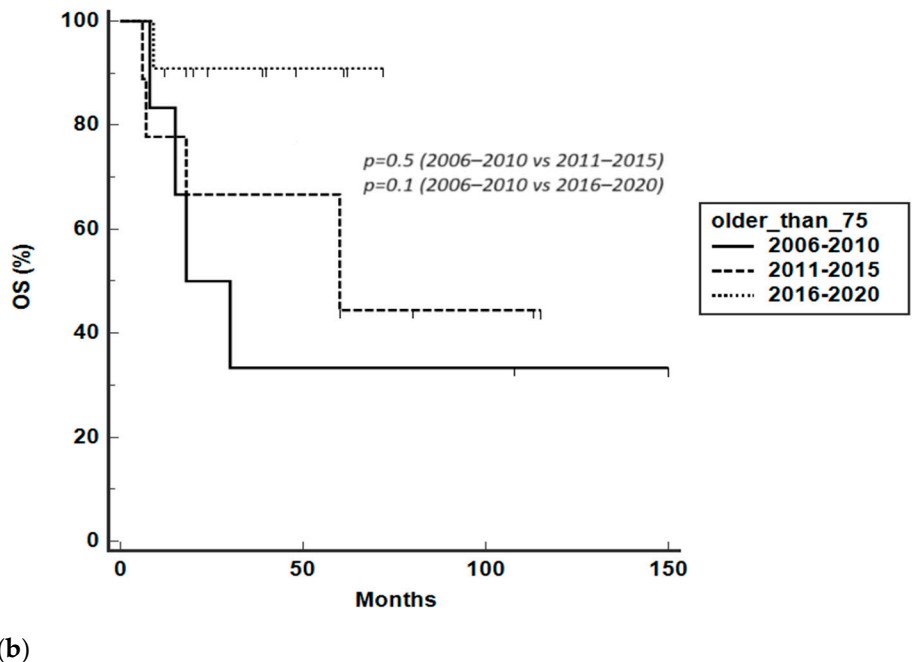

**(b)**

**Figure 4.** PFS (**a**) and OS (**b**) for patients aged 75 years or older according to specified treatment periods (2006–2010, 2011–2015, 2016–2020).

*3.4. Long-Term Toxicity, Second Malignancies, and Causes of Death*

Long-term toxicity, SPM, and causes of death are illustrated in Table 3. There were 25 registered deaths during the long-term follow-up period. The majority of them (12/25; 48%) were due to PD, whereas the remaining were due to SPM (5/25, 20%), infections (4/25, 16%, including one due to COVID-19), unknown origin (3/25, 12%, due to cardiac failure in two elderly patients and a patient with pre-existing cardiac disease in their medical history), and thrombotic thrombocytopenic purpura (1/25, 4%). No other late toxicity was reported as being possibly related to MCL therapy. It should be noted that the SPMs included one case with lung cancer, one with secondary acute myeloid leukemia, one with melanoma, one with non-melanoma skin cancer, and one with biliary tract carcinoma.

**Table 3.** Causes of death and second primary malignancies for the whole cohort.

|  | **Number of Patients (%)** |
| --- | --- |
| Causes of death | 25/73 (34.2%) |
|     Progressive disease | 12/25 (48%) |
|     Second primary malignancies | 5/25 (20%) |
|     Infections | 4/25 (16%) |
|     Unknown origin | 3/25 (12%) |
|     Thrombotic thrombocytopenic purpura | 1/25 (4%) |
| Second primary malignancies | 5/73 (6.8%) |
|     Lung cancer | 1/5 (20%) |
|     Acute myeloid leukemia | 1/5 (20%) |
|     Melanoma | 1/5 (20%) |
|     Non-melanoma skin cancer | 1/5 (20%) |
|     Biliary tract carcinoma | 1/5 (20%) |

## 4. Discussion

We investigated the survival outcome of MCL patients over a 15-year treatment interval, between 2006 and 2020. MCL prognosis has significantly improved in recent years; however, the possible long-term survival benefit of new treatment options has not been clearly established outside of clinical trials [1–4].

According to current guidelines, first-line treatment should include rituximab, HD cytarabine-containing regimens, and ASCT as consolidation for younger, chemosensitive patients, while older patients should receive chemoimmunotherapy, such as BR, R-BAC, R-CHOP, and VR-CAP [1–4]. The recently published, long-term follow-up of the R-BAC regimen in older, newly diagnosed MCL patients demonstrated sustained efficacy over time. After a median follow-up of 86 months, the 7-year PFS and OS were 55% and 63%, respectively, with no relapse reported after the sixth year [11]. The VR-CAP regimen, after a long-term follow-up of 82 months, demonstrated an improved median OS compared to R-CHOP (90·7 vs. 55·7 months, *p* = 0·001). In recent years, rituximab as maintenance treatment has demonstrated OS benefit in both younger and older cases after first-line therapy [10,36].

The management of R/R MCL patients is challenging and possible treatment options include chemoimmunotherapy with a different regimen from that used in front-line treatment, ibrutinib, lenalidomide, temsirolimus, and bortezomib [11,19]. Despite promising results, frequent relapses and an increasingly shorter duration of response have been reported after each line of therapy [11–15]. Ibrutinib represents the most effective novel agent and has shown prolonged efficacy with manageable toxicity [19–23]. In a long-term follow-up analysis across three clinical trials, median PFS was 12.8 months; interestingly, the longest PFS was achieved for patients receiving ibrutinib as second-line therapy [23]. These findings were confirmed in an extended 3.5-year pooled follow-up analysis, in which median PFS and OS for patients receiving ibrutinib in second-line treatment were significantly prolonged compared to those used as third- or later-line treatments (25.4 vs. 10.3 months and not reached vs. 22.5 months, respectively) [37].

Unexpectedly, there are only a few real-life published research papers assessing the long-term survival of MCL patients across multiple years and most of them were assessed before the introduction of rituximab and novel agents [29–31].

Harmanen and colleagues provided a retrospective survival analysis of MCL patients treated between 2000 and 2020 in a population-based cohort of 564 MCL patients; median OS was 80 months, with a 2-year, 5-year, and 10-year OS of 77%, 58%, and 32%, respectively [29]. Remarkably, the entire cohort was divided into three groups according to age at diagnosis; median OS was reduced for patients aged 75 years or older (37 months), compared to patients aged between 65 and 74 years (93 months) and patients younger than 65 years (125 months). When assessing OS in accordance with treatment era (2000–2005, 2006–2010, 2011–2015, and 2016–2020), the authors did not demonstrate any association between OS and a specified time period [29]. The strength of this study is represented by the large sample size, collected outside of clinical trials over 20 years. However, the assessment of PFS is lacking and OS analysis for different age subgroups was not stratified according to treatment periods [29]. In our opinion, PFS is relevant for MCL, which is characterized by multiple relapses across multiple years. Furthermore, the possible OS benefit of new treatment options for patients managed in the last few years should be evaluated, especially for elderly MCL cases. Wu and colleagues reported a significant increase in median OS from 67 to 107 months for patients treated between 2001 and 2012 compared to those treated between 1995 and 2000, while median OS for patients diagnosed between 2013 and 2016 was not reached [30]. The 3-year OS was 67.6%, 72.7%, and 75% for the three different groups, respectively. However, this retrospective study only included younger patients (<65 years old), PFS was not investigated, patients diagnosed after 2016 were not included, and information regarding specific treatment regimens at diagnosis and at relapse was not available [30]. Yang and colleagues investigated 805 Chinese patients between 1999 and 2019, with a median age of 60 years.

In the entire cohort, 5-year PFS and OS were 30.9% and 65%, respectively [31]. Multivariate analysis demonstrated that patients with high–intermediate/high risk MIPI scores, treated without HD cytarabine and/or without maintenance therapy, were associated with reduced OS [31]. The MANTLE-FIRST study evaluated the outcome of 261 MCL cases with R/R disease after HD cytarabine-containing regimens [24]. Ibrutinib and R-BAC were

associated with improved PFS from the time of salvage therapy compared to BR or other; median PFS was 24, 25, 13, and 7 months, respectively. In this study, elderly patients who did not receive HD cytarabine were not included and patients were not divided according to treatment period [24].

Overall, our survival data are comparable with Harmanen and colleagues and we observed a PFS improvement for MCL patients treated after 2010. This finding could be due to the introduction of BR as a first-line regimen for elderly patients, which demonstrated sustained efficacy in both clinical trials and real-life experiences [12,38–42]. In our experience, the BR regimen represents an effective and manageable therapeutic strategy, even for elderly or unfit/frail patients, without unexpected adverse events even after a long-term follow-up period. During recent years, we have witnessed a progressive reduction in the use of R-CHOP or purine analogs. The lack of OS benefit across the years may be due to the fact that OS is influenced by subsequent regimens. Interestingly, the abovementioned OS benefit during the last 5 years for elderly patients could be due to the introduction of ibrutinib as a second-line regimen in Italy, unlike its limited use in Finland, where it is reimbursed as a fourth-line regimen [29,35,43–46]. The main limitation of our study is represented by the retrospective nature of the study and the reduced sample size, and this may explain why some classic prognostic factors, such as MIPI score and stage, did not affect PFS and OS in our cohort. Another limitation is that patients did not receive rituximab as maintenance therapy, which may ensure further PFS and OS improvement in future studies [10,47].

## 5. Conclusions

We found that MCL survival at our center has improved over the years compared to historical cohorts in the pre-rituximab era, even in patients aged 75 years or older, who are not eligible for intensive treatment regimens.

In conclusion, this retrospective analysis in a tertiary care center suggests that MCL prognosis may have been improved since the introduction of the BR regimen as a first-line therapy for elderly patients and novel agents, especially ibrutinib, for R/R cases, regardless of age. In the future, novel strategies, such as CAR T-cell therapy and rituximab as maintenance therapy, could improve MCL survival for newly diagnosed and R/R cases.

**Supplementary Materials:** The following supporting information can be downloaded at https://www.mdpi.com/article/10.3390/hematolrep16010006/s1: Figure S1: Progression-free survival for younger patients <65 years old; Figure S2: Overall survival for younger patients <65 years old; Figure S3: Progression-free survival for patients aged between 65 and 74 years; Figure S4: Overall survival for patients aged between 65 and 74 years; Figure S5: Progression-free survival for patients aged 75 years or older; Figure S6: Overall survival for patients aged 75 years or older.

**Author Contributions:** Conceptualization, E.C. and A.F.; data curation, N.C., M.F., A.D., S.F., B.E.V. and G.L.N.; investigation, N.C., M.F., A.D., S.F., B.E.V. and G.L.N.; methodology, E.C.; supervision, M.B.; writing—original draft, E.C.; writing—review and editing, A.F. and M.B. All authors have read and agreed to the published version of the manuscript.

**Funding:** This research received no external funding.

**Institutional Review Board Statement:** The study was conducted in accordance with the Declaration of Helsinki and approved by the Institutional Review Board (Comitato Etico Regionale per la Sperimentazione Clinica della Regione Toscana Sezione: AREA VASTA SUD EST, protocol approval on 15 October 2023, protocol code SI_MCL, protocol number 25529).

**Informed Consent Statement:** Informed consent was obtained from all of the subjects involved in the study.

**Data Availability Statement:** All data generated or analyzed during this study are included in this article. The original database and further inquiries are available upon request to the corresponding author.

**Conflicts of Interest:** The authors declare no conflicts of interest.

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
