# Peer review of "Survival Outcomes of Patients with Mantle Cell Lymphoma: A Retrospective, 15-Year, Real-Life Study"

_hematolrep, doi:10.3390/hematolrep16010006_

Round 1
Reviewer 1 Report
Comments and Suggestions for Authors
The authors present a very nicely written review of the outcomes of patients treated with Mantle Cell lymphoma at their institution within the last 20 years. While the paper is topical (mantle cell lymphoma continues to be incurable and there is debate about optimal therapy and about the survival of patients today), there are some strong limitations to the manuscript that are not well presented.
The population assessed is NOT a population-based series as is presented (the discussion compares these results to other population-based analyses). These patients were selected to be referred for treatment at this institution. The authors should better describe how patients are selected for their treatment institution in Italy to better understand the population. The authors comment that they included even patients treated with palliation yet their series includes only patients who received active therapy. There were no patients in 20 years who were too frail to be treated at all? No patients with CNS disease who may be rare but do very poorly? No patients with competing comorbidities that would make intensive therapy (rituximab isn't convenient for a frail patient) difficult or not feasible? The manuscript would be improved by better understanding how these patients were selected.
Given that this is a single-centre review for a rare disease, the number of patients is expectedly small. Because of this, conclusions about the results should not be strong. I see no value in performing an analysis for the influence of POD24 for example - if a significant effect was noted, would anyone believe it or care when the sample size in the study is so small?
The paper is well written as a single centre review of a rare lymphoma subtype - it could be simplified (remove any multivariate analyses that are not confident with such small numbers) and the conclusions made more relevant ("the survival at our centre has improved")
Author Response
Response to Reviewer 1
- Reviewer 1 says the paper is nicely written and the topic is very relevant.
We would like to thank the reviewer for this positive comment
- The reviewer says the population assessed is not a population-based series as is presented but the patients were selected to be referred for treatment at our Institution.
We agree with the reviewer, our study is a single-center experience. In the Methods section, we did not describe our study as a population-based study, but we included all MCL patients treated in 15 years at our Institution. In the Discussion, we included the comparison with the population-based study by Harmanen and colleagues, due to there are very few real-life published experiences with long-term follow-up. We have removed the term population-based in the text in the Discussion, line 262.
-The reviewer suggests to better describe how patients were selected for treatment.
We agree with the reviewer, we have expanded the description of inclusion criteria, Methods Section, page 3, lines 92-95, as showed below.
“Patients were included if they had a MCL diagnosis according to WHO classification, were≥18 years old at diagnosis, received at least 1 cycle of systemic therapy according to the prescribing information approved by the “Agenzia Italiana del Farmaco” and there were available data regarding treatment response and long-term toxicity.”
Moreover, regarding treatment criteria, we used ESMO guidelines since 2013 and expert recommendations in previous years. Since MCL is an aggressive disease, we treat all patients at diagnosis except those with leukemic non-nodal MCL. We have added this information, Methods Section, page 3, lines 96-100, as showed below.
“As regards the treatment criteria, at our Institution we used the ESMO guidelines since 2013, the year in which they were published [4,33]. In previous years, since MCL therapy was not standardized, we adopted expert recommendations [34]. Overall, due to MCL is an aggressive disease, our policy was to treat all patients at diagnosis, except those with a LNN presentation.”
The reviewer says that we included even patients treated with palliation yet our series includes only patients who received active therapy. There were no patients in 20 years who were too frail to be treated at all?
We partially agree with the reviewer. We believe as treatment policy at our Institution that there are no patients too frail to receive at least an alkylating agent or low-dose fludarabine, with or without rituximab.
The reviewer says if there were patients with CNS disease.
We agree with the reviewer, no patients had CNS disease at diagnosis but 2 cases had CNS relapse. We have added this information, Results section, page 5, paragraph 3.2 lines 161-163, as showed below.
“Interestingly, two cases experienced central nervous system relapse and received ibrutinib or chemoimmunotherapy (1 case each).”
The reviewer says if there were patients with comorbidities that would make intensive therapy not feasible and that rituximab isn't convenient for a frail patient.
We partially agree with the reviewer. We believe that rituximab, as for DLBCL, should not be denied due to age or frailty, but should be carefully administered according to data sheet. In our cohort, only 1 case did not receive rituximab, who presented with severe cardiac failure (New York Heart Association score was 3). Overall, we considered as palliation the 4 cases who received alkylators, with or without rituximab, and the case treated with low-dose, oral fludarabine and rituximab. We have added this information in the Results Section, paragraph 3.2, lines 154-158, as showed below.
“Remarkably, in our cohort, only 1 case did not receive rituximab, who presented with severe cardiac failure (New York Heart Association score was 3). Overall, we considered as palliation the 4 cases who received alkylating agents, with or without rituximab, and the case treated with low-dose, oral fludarabine and rituximab.”
The reviewer says, given that this is a single-centre review for a rare disease, the number of patients is expectedly small. Because of this, conclusions about the results should not be strong. The reviewer identifies no value in performing an analysis for the influence of POD24 for example.
We agree with the reviewer and we have removed POD24 analysis in the Methods and in the Results section. Moreover, we have modified the conclusion by using weaker terms, as showed below.
“In conclusion, this retrospective analysis in a tertiary care center suggests that MCL prognosis may have been improved since the introduction of BR regimen as first-line therapy for elderly patients and novel agents, especially ibrutinib, for R/R cases, regardless of age.”
The paper is well written as a single centre review of a rare lymphoma subtype - it could be simplified (remove any multivariate analyses that are not confident with such small numbers) and the conclusions made more relevant ("the survival at our centre has improved").
We agree with the reviewer, we removed multivariate analysis and the conclusion has been modified, as showed below.
“We found that MCL survival at our centre has improved during years…”
Reviewer 2 Report
Comments and Suggestions for Authors
In this manuscript, the authors retrospectively investigated the long-term efficacy and toxicity of 73 MCL patients. This report is interesting but has some issued to be addressed.
1. In terms of the Result 3.4, I would like to suggest authors to make a new table to summarize the long-term toxicity, second malignancies and causes of death. It will make readers much easier to get the information.
2. The p value of 0.04 in Figure 2 should be indicated in detail, for example, between which two groups; the p values in Figure 3 and Figure 4 should be indicated as well even if no statistical significance were reached.
3. English needs to be carefully edited due to many grammatical errors and typos, for example the past tense should be used when describing the results in Abstract (e.g., “we reported” rather than “we report”), and the same is true in Result 3.3; in Introduction, “many elderly MCL patients are not included” should be “…were not included”; in Result 3.1, “the entire cohort and by treatment era” should be “the entire cohort by treatment era”; in Discussion, the last paragraph, “our chort” should be “our cohort”, etc.
Comments on the Quality of English Language
Extensive editing of English language is required due to many grammatical errors and typos.
Author Response
Response to Reviewer 2
The reviewer says the report is interesting but has some issues to be addressed.
First of all, we would like to thank the reviewer for the positive comment.
1.In terms of the Result 3.4, the reviewer suggests to make a new table to summarize the long-term toxicity, second malignancies and causes of death.
We agree with the reviewer, we have added Table 3, as showed below.
Table 3. Causes of death and second primary malignancies for the whole cohort
|
Number of patients (%) |
Causes of death Progressive disease Second primary malignancies Infections Unknown origin Thrombotic thombocytopenic purpura Second primary malignancies Lung cancer Acute myeloid leukemia Melanoma Non-melanoma skin cancer Biliary tract carcinoma |
25/73 (34.2%) 12/25 (48%) 5/25 (20%) 4/25 (16%) 3/25 (12%) 1/25 (4%) 5/73 (6.8%) 1/5 (20%) 1/5 (20%) 1/5 (20%) 1/5 (20%) 1/5 (20%) |
2.The reviewer says the p value of 0.04 in Figure 2 should be indicated in detail, for example, between which two groups.
We agree with the reviewer, we have updated Figure 2, as requested.
The reviewer says p values should be indicated in Figures 3 and 4, even if no statistical significance was reached.
We agree with the reviewer, we have updated Figures 3 and 4, as requested.
3.The reviewer says English language should be improved, due to grammatical errors and typos.
We agree with the reviewer and we apologize for the inconvenience.
As requested, we used “we reported” rather than “we report” in the Abstract and in the results, “elderly MCL patients were not included” rather than “are not included”, “our cohort” rather than “our chort” in the last paragraph.
Introduction, line 35, “during” was changed to “over the”. Lines 39-40, the sentence has been modified, as showed below
“However, 10-20% of patients are characterized by a leukemic, non-nodal (LNN) presentation, with a more indolent clinical behavior and prolonged survival [4-6].”
Line 53, “better” was changed to “more efficacious”.
Lines 56 and 64, “further” has been changed to “subsequent”.
Lines 63-65, the sentence was modified, as showed below.
“For patients failing ibrutinib, overall prognosis is extremely poor after ibrutinib failure, with an unsatisfactory response to subsequent regimens [26].”
Line 67, “awaiting” has been changed to “waiting for”.
Materials and methods, lines 78-79 the sentence was modified, as follows.
“From 2006 to 2020, 73 consecutive newly diagnosed or R/R MCL patients were enrolled in this observational, real-world monocenter study.”
Lines 92-95, the sentence about inclusion criteria was modified, as showed below.
“Patients were included if they had a MCL diagnosis according to WHO classification, were≥18 years old at diagnosis, received at least 1 cycle of systemic therapy according to the prescribing information approved by the “Agenzia Italiana del Farmaco” and there were available data regarding treatment response and long-term toxicity.”
Line 110, “none of the enrolled patients” was changed to “none of them”.
Lines 115-116, “The detailed later” was changed to “Details regarding subsequent treatment strategies”.
Line 114-115, “administered in the indications” was changed to “administered following the indications” and “later treatment strategies” was changed to “subsequent treatment strategies”.
Line 130, “survival analysis were assessed” was changed to “Survival analyses were assessed”.
Results, line 148, the words “of which” were added, as showed below.
“…21/24 of which responded to treatment and underwent ASCT)…”
Line 147, the sentence was changed, as showed below.
“First-line regimens, as represented in Table 2, included…”
Line 164, “later” was changed to “subsequent”.
Lines 149 and 173,”doxorubicine” was changed to “doxorubicin”.
Line 176, “entire” was changed to “whole”.
Line 214, “showed” was changed to “observed that”.
Line 271, “youger” was changed to “younger”.
Line 307, “that no patients received rituximab” was changed to “that patients did not receive rituximab”
Reviewer 3 Report
Comments and Suggestions for Authors
The authors report on the results of mantle cell lymphoma treatment experienced at their institution for 15 years.
As the authors point out, new drugs are being introduced every year and survival rates are expected to improve, the real-world outcome is a topic of great interest.
I have some points that I would like to confirm with the authors.
It is stated that the study underwent ethical review on October 15, 2023, and that written consent was obtained from all patients. It is unclear how consent was obtained for participants who died.
The study does not include patients who received rituximab maintenance therapy. Why?
The study compares outcomes by decade, but does not indicate the duration of observation for each.
It is stated that the introduction of bendamustine may have improved progression-free survival, but it is unclear how many patients received bendamustine and in which decade.
I would like to see data on the response rate after 1st line treatment.
The background of patients in each age group seems to be generally the same, but there are only a small number of patients in each group, so statistical comparison is not meaningful.
It is stated that MIPI and stage did not affect progression-free survival and overall survival, but this may be a sample size issue.
Author Response
Response to Reviewer3
The reviewer says, due to new drugs are being introduced every year and survival rates are expected to improve, the real-world outcome is a relevant topic for MCL.
We would like to thank the reviewer for the positive comment.
The reviewer says it is unclear how consent was obtained for participants who died.
We agree with the reviewer. Informed consent was signed by the patients who were alive at the time of ethical approval. For participants who died, the consent was considered as acquired by Italian law, according to the Authorization n.9/2016- General authorization to process personal data for scientific research (date of authorization was 15-dec-2016).
We have added this information in the Methods Section, lines 86-89.
The reviewer asks to explain why the study does not include patients who received rituximab maintenance.
We agree with the reviewer, it was due to the fact that rituximab, as maintenance therapy after first-line regimen for MCL patients, was included in the list of reimbursed drugs, according to Italian law no. 648/1996, on 22-may-2023. Prior to this date, we have not administered rituximab as maintenance therapy to any patients. We have added this information, Methods Section, lines 110-113.
The reviewer asks to indicate the duration of observation for the different treatment periods.
We agree with the reviewer. We have added this information, Results section, lines 112-113, as showed below.
“The median duration of observation for the different treatment periods was 36 months (range 6-180 months), 42 months (range 6-120 months) and 24 months (range 6-72 months), respectively.”
The reviewer says the introduction of bendamustine may have improved PFS, but it is unclear how many patients received bendamustine and in which decade.
We partially agree with the reviewer, the total number of patients receiving bendamustine with rituximab is 26/73, while 9 cases received R-BAC, as indicated in Table2. Bendamustine was administered since 2013 at our Institution (the year of approval in Italy as first-line regimen). We have added the total number of patients receiving bendamustine and in which decade, Results section, paragraph 3.2, lines 152-154, as showed below.
“Total number of patients who received bendamustine was 35/73 (47.9%); it was administered to 16/23 patients (69.6%) and to 19/31 patients (61.3%) between 2011-2015 and 2016-2020, respectively.”
The reviewer says to show the response rate after first-line treatment.
We have added this information, Results Section, paragraph 3.2, lines 158-159. We have not analyzed the different regimens or treatment periods due to the reduced sample size.
“Treatment response was CR for 51/73 cases (69.8%), partial response for 10/73 cases (13.7%) and stable disease/PD for 12/73 cases (16.5%), respectively.”
The reviewer saysthe background of patients in eaxch age group is similar, but statistical comparison is not meaningful due to the reduced sample size.
We partially agree with the reviewer. We have reported the reduced sample size as one of the study limitations. However, statistical comparison could help the reader and support the rationale of an improved survival, due to the introduction of new drugs being during years.
The reviewer says MIPI score and stage did not affect survival, but this may be due to reduced sample size.
We agree with the reviewer and we have explained study limitations in the Discussion, lines 301-305, as showed below.
“The main limitation of our study is the reduced sample size, and this may explain why some classic prognostic factors, such as MIPI score and stage, did not affect PFS and OS in our cohort. Another limitation is that patients did not receive rituximab as maintenance therapy, which may ensure a further PFS and OS improvement in future studies [10,47].”
Round 2
Reviewer 1 Report
Comments and Suggestions for Authors
The authors have made substantial improvements to the manuscript based on the 3 reviewers comments. All appropriate revisions have been completed.
Author Response
We would like to thank the reviewer for the positive comments
Reviewer 3 Report
Comments and Suggestions for Authors
The authors carefully added appropriate information in response to the reviewers' comments, making the paper more useful to readers. If possible, I would like the authors to revise Table 2 as well as Table 1 in the total number column plus the plus the decade group column.
Author Response
The reviewer says the authors carefully added appropriate information in response to the comments, making the paper more useful to readers.
We would like to thank the reviewer for the positive comments.
The reviewer says to to revise Table 2 as well as Table 1 in the total number column plus the decade group column.
We agree with the reviewer, we have updated Table 2 in the text, as showed below .
Table 2. First-line regimens for the entire cohort
Treatment |
Number of patients (%) |
2006-2010 n=19 |
2011-2015 n=23 |
2016-2020 n=31 |
Rituximab-bendamustine High-dose cytarabine-based therapies Autologous stem-cell transplantation Rituximab-bendamustine-cytarabine (R-BAC) Rituximab-CHOP or CHOP-like regimens Fludarabine-containing regimens Rituximab-alkylating agents Alkylating agents |
26/73 (35.6%) 24/73 (32.9%) 21/73 (28.8%) 9/73 (12.3%) 6/73 (8.2%) 5/73 (6.8%) 2/73 (2.8%) 1/73 (1.4%) |
0 6/19 (31.6%) 6/19 (31.6%) 0 5/19 (26.3%) 5/19 (26.3%) 2/19 (10.5%) 1/19 (5.3%) |
9/23 (39.1%) 8/23 (34.8%) 6/23 (26%) 5/23 (21.7%) 1/23 (4.4%) 0 0 0 |
17/31 (54.8%) 10/31 (32.3%) 9/31 (29%) 4/31 (12.9%) 0 0 0 0 |